# Fucoidan from *Lessonia trabeculata* Induces Apoptosis through Caspase Dependent and Caspase-Independent Activation in 4T1 Breast Adenocarcinoma In Vitro

**DOI:** 10.3390/md22060251

**Published:** 2024-05-29

**Authors:** Raisa Teresa Cruz Riquelme, Erasmo Honorio Colona-Vallejos, Libertad Alzamora-Gonzales, Rosa María Condori Macuri

**Affiliations:** Research Group Immunomodulators and Antitumor of Natural and Synthetic Origen, Immunology Laboratory, Universidad Nacional Mayor de San Marcos, Lima 11-0058, Peru; lalzamorag@unmsm.edu.pe (L.A.-G.); rosa.condori@unmsm.edu.pe (R.M.C.M.)

**Keywords:** *Lessonia trabeculata*, fucoidan, cytotoxicity, antiapoptotic, proapoptotic, breast cancer

## Abstract

Experiments conducted on triple-negative breast cancer have shown that fucoidan from *Lessonia trabeculata* (FLt) exhibits cytotoxic and antitumor properties. However, further research is necessary to gain a complete understanding of its bioactivity and level of cytotoxicity. The cytotoxic effect of FLt was determined by the 2,5-diphenyl-2H-tetrazolium bromide (MTT) assay. Apoptosis was analyzed using annexin V and caspase 3/7 staining kit and DNA fragmentation. In addition, transcriptional expression of antiapoptotic (Bcl-2 and XIAP) and proapoptotic (caspase 8, caspase 9, and AIF) genes were analyzed in TNBC 4T1 cells. After 72 h of culture, the IC_50_ for FLt was 561 μg/mL, while doxorubicin (Dox) had an IC_50_ of 0.04 μg/mL. In addition, assays for FLt + Dox were performed. Annexin V and caspase 3/7 revealed that FLt induces early and late-stage apoptosis. DNA fragmentation results support necrotic death of 4T1 cells. Similarly, transcripts that prevent cell death were decreased, while transcripts that promote cell death were increased. This study showed that FLt induces apoptosis by both caspase-dependent and caspase-independent mechanisms. These findings suggest that FLt may have potential applications in breast cancer treatment. Further research will provide more information to elucidate the mechanism of action of FLt.

## 1. Introduction

Breast cancer (BC) is the neoplasm with the highest incidence and mortality, a trend that continues to increase due to population growth, aging, and lifestyle choices [1]. Worldwide, BC has become the most common neoplasm, surpassing lung, colorectal, prostate, and stomach cancers [2]. By 2040, 3.19 million new cases and 1.04 million deaths are expected [3]. In Peru, the estimated incidence and mortality rates by 2040 are 10.5 thousand and 3.09 thousand, respectively [4].

Despite the current availability of a variety of therapies for BC, the complexity of this disease and the limited accessibility of these treatments due to the personalized therapeutic approach pose a challenge in finding effective therapies, particularly for patients with triple-negative breast cancer (TNBC). TNBC patients have the worst prognosis and lower survival rates [5].

In this scenario, marine macroalgae serve as a source of molecules with potential pharmaceutical and nutraceutical benefits for human health due to their array of polysaccharides [6]. Among these polysaccharides, fucoidans exhibit various biological activities, such as anticancer, antioxidant, antiviral, antithrombotic, and anti-inflammatory effects. These activities depend on the structural complexity and heterogeneity of their composition, the degree of sulfation, monosaccharide composition, type and position of substituents, species, location, season, and cultivation parameters [7].

Caspase-dependent apoptotic pathways can be endogenous, originating from mitochondria or the endoplasmic reticulum, or exogenous, originating from death receptors [8]. In addition, there is a caspase-independent pathway in which apoptosis-inducing factor (AIF) is critical for poly (ADP-ribose) polymerase (PARP)-dependent necrotic cell death [9].

The apoptotic effect of fucoidan has been demonstrated in vitro, showing activation of caspases and PARP in human colon cancer HT-29 cells [10] and human pancreatic adenocarcinoma cells anc-1, MiaPaCa-2, Panc-3.27 and BxPC-3 [11]. These include regulation of caspases at the transcriptional and translational levels in human BC MCF-7 cells [12], positive regulation of apoptotic markers (Bax, PARP, and caspase 3), negative regulation of antiapoptotic markers (Bcl-2) and nuclear fragmentation in CL-6 cholangiocarcinoma cells [13].

However, due to the complexity and diversity of fucoidans within brown algae species, it is not possible to define an equal or similar effect [14], so the apoptotic effect of fucoidan from *Lessonia trabeculata* (FLt) from our coastal regions on cancer cells is being investigated.

Previous research has determined the immunomodulatory effect of FLt on human peripheral blood mononuclear cells [15], as well as its cytotoxic and apoptotic activity on human squamous cell carcinoma type 2 (Hep-2) [16], cervical adenocarcinoma (HeLa), and human promonocytic leukemia (U937) cell lines [17]. However, further research is needed to understand the additional bioactivities that FLt may have on a variety of human tumor cells.

This study demonstrated in vitro that FLt has high cytotoxicity and selectivity and induces apoptosis through activation of caspase 3/7 and transcriptional expression of proapoptotic genes (caspase 8 and caspase 9) in TNBC 4T1 cells. Fucoidan inhibited the expression of antiapoptotic transcripts (Bcl-2 and XIAP). In addition, transcriptional expression of AIF could induce apoptosis by the intrinsic caspase-independent pathway and, consequently, the necrotic activity shown by the DNA smear. Further research is needed to clarify the hypothesized mechanism of induction of apoptotic pathways.

## 2. Results and Discussion

### 2.1. Effect of FLt on Cell Viability in 4T1 Cell Line

After 72 h of FLt treatment, the percentage of inhibition of 4T1 cells was significant in the range of 100 to 10,000 μg/mL compared to NC (*p* < 0.001) (Figure 1A). At 1000 μg/mL FLt, growth inhibition was observed in the 4T1 (48.8%) and VERO-76 (22%) cell lines (*p* < 0.001). Similarly, Toccas-Salas et al. found a 56% inhibition for the 4T1 cell line with a FLt-rich extract [18]. Dox showed higher cytotoxic activity for both cell lines (*p* < 0.05) (Figure 1B). FLt exerts more potent cytotoxic effects on the 4T1 cell line than on VERO-76 cells in a concentration-dependent manner.

### 2.2. Half-Maximal Inhibitory Concentration (IC_50_) of FLt and Dox

The IC_50_ of FLt and Dox for 4T1 cells was 561 μg/mL and 0.04 μg/mL, respectively, while for the diploid VERO-76 line, it was 6823 μg/mL and 0.2 μg/mL, respectively. The SI of FLt and Dox were 12.2 and 5, respectively, demonstrating high selectivity against cancer cells when values are higher than 1 [19].

### 2.3. Analysis of Cell Membrane Phosphatidylserine Externalization (Annexin V)

Annexin V binding to phosphatidylserine (PS) was analyzed to determine the percentage of apoptotic cells (Figure 2A,B). The IC_50_ of FLt induced a total apoptotic effect (47.8%, *p* < 0.001) compared to NC, while the IC_50_ of FLt + Dox (40.8%) showed no significant difference compared to IC_50_ of FLt. In addition, FLt increased the percentage of early (19.53%) and late/dead (28.30%) apoptosis compared to IC_50_ of FLt + Dox (13.12%). Similarly, treatment with crude fucoidan extract from *Fucus vesiculosus* produces a significant early apoptotic effect in 4T1 cells [20], while fucoidan from *Sargassum aquifolium* decreases the percentage of live cells and increases late apoptotic cells in the human lung carcinoma A-549 cell line [21]. In this study, the inhibition of cell proliferation would be associated with the apoptotic activity of fucoidan.

### 2.4. Induction of Caspase 3/7 Activation in 4T1 Cells

The IC_50_ of FLt induced the activation of caspase 3/7 (60.6%, *p* < 0.001) compared to the NC (Figure 2B,D). Similarly, Teruya et al. determined that fucoidan from *Cladosiphon okamuranus* TOKIDA induced apoptosis through the activation of caspase 3/7 in U937 cells [22]. It has been reported that fucoidan-induced apoptosis in MCF-7 cells requires caspase 7 to the exclusion of caspase 3 [23], so it would be necessary to evaluate these independently. However, fucoidans can induce apoptosis through the specific activation of caspases such as caspase 3, 7, 8, and 9 in various tumor cell lines [24]. Although the IC_50_ of FLt + Dox increased caspase 3/7 activity (88.7%), it did not exceed the effect of the IC_50_ of Dox (94.2%), indicating that the influence of FLt in FLt + Dox would not further enhance caspase 3/7 induction (Figure 2B,D). These results suggest that FLt induces the activation of effector caspases.

### 2.5. Effect of FLt on DNA Integrity

The DNA fragmentation assay demonstrated the enhanced anticancer activity of FLt in 4T1 cells. Treatment with FLt IC50 for 72 h resulted in the formation of visible length fragments of sheared DNA as smears compared to NC. In addition, the same effect was observed with the IC_50_ of FLt + Dox and only with the IC_50_ of Dox (Figure 3). Similarly, Condori et al. observed that FLt treatment of spheroids formed from 4T1 cells plus mouse splenocytes produced necrotic areas [25]. Treatment of the liver cancer cell line HepG2 with *Stoechospermum marginatum* fucoidan for 24 h has been shown to induce necrotic activity [26]. In addition, the high percentage of cells in apoptosis/late death observed with annexin V and caspase 3/7 assays (Figure 2A,C) would also support the necrotic effect of FLt. The observation of fewer or more DNA ladders and DNA smears as a product of DNA fragmentation with an internucleosomal or random pattern, respectively, would depend on the time and inducer used [27]. Based on the DNA fragmentation assay, it is suggested that treatment of 4T1 cells with FLt has anticancer activity based on the observation of the DNA smear. However, complementary assays are required to define a possible programmed cell death called necroptosis.

### 2.6. Transcriptional Expression (mRNA) of Antiapoptotic and Proapoptotic Genes

The mRNA expression of antiapoptotic factors (Bcl-2 and XIAP) and proapoptotic factors (caspase 8, caspase 9, and AIF) in 4T1 cells treated with the IC_50_ of FLt, Dox, and FLt + Dox is shown in Figure 4. Bcl-2 gene transcript was not expressed with the IC_50_ of FLt, Dox, and FLt + Dox compared to NC, while XIAP mRNA was only expressed with the IC_50_ of FLt + Dox (Figure 4A,B). Similar results were obtained by Xue et al., who found that commercial *Fucus vesiculosus* fucoidan inhibited Bcl-2 expression and enhanced caspase 3 activation in 4T1 cells [20]. Similarly, commercial *Fucus vesiculosus* fucoidan decreased the transcriptional expression of Bcl-2 in a concentration-dependent manner on the HCC hepatocellular carcinoma cell line, thereby promoting apoptosis [28].

Regarding the XIAP protein, it has been reported to exert potent antiapoptotic effects by inhibiting caspase 3 and 9 [29] and mediating proinflammatory signaling [30]. Thus, FLt could be used as a promising treatment for cancer and inflammatory diseases [31]. However, the expression of XIAP in FLt + Dox treatment may regulate necroptosis [29], which needs to be confirmed in future investigations. Banafa et al. showed that commercial fucoidan decreased the transcriptional expression of XIAP in MDA-MB 231 BC cells while increasing the expression of caspase 8, 3, and 9 [32]. Yang et al. found that *Undaria pinnatifida* fucoidan downregulated XIAP expression and induced caspase 3 and 9 activation [33], as observed in this study. Although FLt + Dox insignificantly induced XIAP expression in 4T1 cells, this effect may be due to the antiapoptotic agent Che-1 (RNA polymerase II binding protein), which activates XIAP expression in response to DNA damage caused by Dox concentration [31], thereby interfering with the apoptotic process.

The mRNA levels of proapoptotic genes treated with the IC_50_ of FLt and Dox were increased compared to NC (*p* < 0.001). The transcript expression of caspase 8 showed no differences between the IC_50_ of FLt, Dox, and FLt + Dox treatments (Figure 4C,D). Similar results were obtained with commercial fucoidan, which upregulates the transcriptional expression of caspase 8 in MCF-7 cells [12]. The mRNA levels of caspase 9 and AIF were higher in the IC_50_ of FLt and Dox treatments than in the IC_50_ of FLt + Dox (*p* < 0.001) (Figure 4C,D). It is noteworthy that FLt + Dox did not increase the expression of proapoptotic proteins, in agreement with Malhão et al. who reported that the combination of fucoidan with Dox had a lower effect than Dox treatment alone [34]. In addition, prolonged exposure to Dox (≥24 h) has been shown to reduce DNA damage in in vitro and in vivo models of BC [34,35]. However, Abudabbus et al. demonstrated greater efficacy in the combined treatment of fucoidan with Dox on BC cell lines [36]. Previous studies have shown that FLt has a modulatory effect on the production of reactive oxygen species [15] and cytokines [25]. Therefore, FLt + Dox could exert a cytoprotective effect on 4T1 by reducing the oxidative and inflammatory effects of Dox [37]. The use of higher concentrations of fucoidan (mg/mL) could improve the cytotoxic effects on BC cell lines [34,38]. However, the anticancer activity of fucoidan depends on the brown alga species, extraction and purification procedures, composition and chemical structure, molecular weight, sulfate content, and other factors [39]. Furthermore, the loss of Dox efficacy would depend on the concentration used in the FLt + Dox treatment and the heterogeneity of the cell line [34].

In this study, the IC_50_ of FLt and Dox treatments increased AIF mRNA expression (Figure 4C,D), which would be related to both the induction of apoptosis and caspase-independent necrosis [40].

The results of this study suggest that FLt induces activation of both the intrinsic and extrinsic pathways of apoptosis in TNBC 4T1 cells by inhibiting the transcriptional expression of Bcl-2 and XIAP. This inhibition would increase caspase 9 expression, activate caspase 3/7, and induce apoptosis via the intrinsic pathway. In addition, expression of the AIF transcript would be associated with necrotic activity. FLt would also increase the expression of caspase 8, which could activate proapoptotic proteins and contribute to the intrinsic pathway or activate caspase 3/7 and enhance the apoptotic process through the extrinsic pathway (Figure 1). These preliminary results suggest that FLt is a promising candidate for adjuvant treatment of TNBC; however, this is a hypothesis that requires further research.

## 3. Materials and Methods

### 3.1. Materials

Fucoidan was provided by Peruvian Seaweeds S.A (PSW S.A, Lima, Peru). Doxorubicin hydrochloride (Dox) was purchased from Sigma Aldrich (St. Louis, MO, USA, cat# D1515). RPMI-1640 was obtained from Sigma Aldrich (St. Louis, MO, USA, cat# R8005), and sodium bicarbonate was obtained from Sigma Aldrich (St. Louis, MO, USA, cat# S5761). Fetal bovine serum (FBS) was sourced from Biowest (Bradenton, FL, USA, cat# S1620), and penicillin–streptomycin solution was obtained from Sigma Aldrich (St. Louis, MO, USA, cat# P4333). Fungizone was acquired from Gibco (Paris, France, cat# 15290018), and Tripsin-EDTA was obtained from Pan Biotech (Aidenbach, Germany, cat# P10-023500). Dimethyl sulfoxide (DMSO, cat# 472301) and 3-(4,5-dimethyl-2-thiazolyl)-2,5-diphenyl-2H-tetrazolium bromide (MTT, cat# M5655) were purchased from Sigma Aldrich (USA). Other chemicals were of analytical grade. PureLink RNA Mini Kit (Invitrogen, Carlsbad, CA, USA, cat# 12183018A), qScript^®^ cDNA SuperMix from Quanta bio (Beverly, MA, USA, cat# 95048-100), Platinum PCR SuperMix (Invitrogen, São Paulo, Brazil, cat# 11306161), SYBR Green (Invitrogen, Carlsbad, CA, USA, ref# S33102), Annexin V Dead Cell Kit Muse™ from Millipore (Burlington, MA, USA, cat# MCH100105), and Muse^®^ Caspase 3/7 kit were obtained from Luminex (Austin, TX, USA, cat# MCH100108).

### 3.2. Cell Line

The mouse triple-negative mammary tumor cell line 4T1 (TNBC, code BCRJ0022) was obtained from the Rio de Janeiro Cell Bank (Brazil). The 4T1 cells were cultured in RPMI-1640 (Sigma Aldrich, USA), supplemented with 10% fetal bovine serum (FBS, Biowest), 0.2% sodium bicarbonate, 1% sodium pyruvate, 1% penicillin–streptomycin, and 0.05% amphotericin (coRPMI) at 37 °C, 5% CO_2_ and 95% humidity.

### 3.3. Fucoidan Preparation

Peruvian Seaweeds (PSW S.A.) isolated FLt from *L. trabeculata* specimens collected in San Nicolás Bay (15°15′39″ S and 75°13′47″ W), Marcona District, Nazca Province, Ica Region, Peru. The FLt obtained by PSW S.A. had a purity of 83.4%, and its characterization determined that 59% of total sugars expressed as galactose and fucose and 5.7% of sulfates [25]. Dilutions of 1, 10, 100, 1000, 1000, 2000, 6000, 8000, and 10,000 μg/mL in coRPMI were prepared from the FLt solution (10 mg/mL) to assess cell cytotoxicity and determine the half-maximal inhibitory concentration (IC_50_).

### 3.4. Cell Viability Assay

The MTT assay was used to assess the cell viability of FLt and doxorubicin (Dox). After being cultured for 72 h in 96-well plates, 4T1 cells (2 × 10^3^ cells/well) and VERO 76 cells (2 × 10^4^ cells/well, normal cell control) were treated with the described concentrations. After exposure to FLt and Dox, the cells were incubated at 37 °C, 5% CO_2_, and 95% humidity. Dox was chosen as a positive control due to its widespread use in antitumor therapy, with concentrations of 0.01, 0.1, 1, 5, and 10 μg/mL. Untreated cells incubated in coRPMI were used as the negative control (NC). A final concentration of 50 μg/mL MTT was added to each well and incubated for 4 h. Absorbances were measured at wavelengths of 570 nm and 630 nm as a reference using a Biotek EPOCH2 spectrophotometer. The percentage of viable cells was calculated as (%) = [100 × (sample abs)/(control abs)].

### 3.5. Determination of Half-Maximal Inhibitory Concentration (IC_50_) and Selectivity Index (SI)

The half-maximal inhibitory concentration (IC_50_) and selectivity index [18] for FLt and Dox were calculated from the data obtained. IC_50_ values were determined by sigmoidal fitting of the data in the GraphPad Prism statistical program (GraphPad software, version 8.01, USA). To assess the apoptotic effect (annexin V, caspase 3/7, DNA fragmentation, and transcriptional expression of antiapoptotic and proapoptotic genes), 4T1 cultures were exposed to the IC_50_ of FLt (561 μg/mL) and Dox (0.04 μg/mL). In addition, an assay using the IC_50_ of FLt and Dox (FLt + Dox) was performed.

### 3.6. Detection of Apoptosis by Annexin V

The Muse Annexin V and Dead Cell Kit (Millipore Co., Burlington, MA, USA) were used for the apoptosis assay. The 4T1 cells (5 × 10^3^ cells/well) were cultured in 24-well plates. After treatment with the IC_50_ described in (item 3.5), the monolayers were trypsinized for 1 min and centrifuged at 1500 rpm, 10 °C for 5 min. The pellet was resuspended in 500 µL of PBS containing 1% FBS. Then, 100 µL of cells were mixed with 100 µL of annexin V reagent and incubated for 20 min at room temperature in the dark. The stained samples were analyzed using the Muse™ Cell Analyzer (Millipore, USA).

### 3.7. Caspase 3/7 Assay

The Muse Caspase 3/7 kit (Luminex Co., Austin, TX, USA) was used for the apoptosis assay. The 4T1 cells (5 × 10^3^ cells/well) were treated with the IC_50_ as described in (item 3.5). The pellet was resuspended in 200 µL of PBS containing 1% FBS. A total of 50 µL of the sample was mixed with 5 µL of the caspase 3/7 reagent for 30 min at 37 °C in the dark. Then, 150 µL of 7-aminoactinomycin D (7-AAD) solution was added, and samples were analyzed on the Muse™ Cell Analyzer.

### 3.8. Determination of DNA Fragmentation

The 4T1 cells (2 × 10^5^ cells/well) treated with the IC_50_ as described in (item 3.5) were washed and resuspended in lysis buffer (2 mM ethylenediaminetetraacetic acid, 10 mM Tris-HCl, 10 mM potassium chloride, 10 mM magnesium chloride) according to [41,42,43,44] with slight modifications. The pellet was resuspended in lysis buffer, 10% SDS, RNase (10 mg/mL), and Proteinase K (20 mg/mL) and incubated at 55 °C and 37 °C for 10 min. Then, 5 M NaCl was added for 5 min at room temperature, and cold absolute ethanol was added to the supernatant for 10 min. The pellet was resuspended in 70% cold ethanol for 10 min and air dried at room temperature. Total DNA, maintained in molecular grade water (Himedia^®^, Mumbai, India), was diluted with SYBR Green dye (Invitrogen™, Carlsbad, CA, USA) and DNA loading buffer (6×) for analysis by 1.7% agarose gel electrophoresis at 80 V for 90 min. DNA integrity was assessed using an Enduro™ GDS gel documentation system (Labnet, NJ, USA).

### 3.9. Analysis of Transcriptional Expression of Bcl-2, XIAP, Caspase 8, Caspase 9, and AIF

The RNA sample from 4T1 cells (5 × 10^3^ cells/well) treated with the IC_50_ as described in (item 3.5) was isolated using the PureLink RNA Mini Kit. cDNA was synthesized according to the specifications of the qScript cDNA SuperMix kit. PCR was performed in a thermal cycler (Blue-Ray Biotech, New Taipei City, China) using standard protocols. For cDNA synthesis, the following protocol was used: 25 °C for 5 min, 42 °C for 30 min, and 85 °C for 5 min. Platinum PCR SuperMix was used for cDNA amplification, with an initial denaturation (94 °C, 5 min), denaturation (94 °C, 30 s), annealing for 30 s, extension (72 °C, 45 s), repeated for 35–40 cycles, and final extension (72 °C, 7 min). The sequences of the primers used to analyze the genes are shown in Table 1.

Amplicons were determined by 1.5% agarose gel electrophoresis (100 V, 60 min), photographed using gel documentation system software, and quantified by densitometry using ImageJ 1.49i software. β-Actin was used as an endogenous control.

### 3.10. Statistical Analysis

Cell viability results are expressed as mean ± standard deviation (S.D.). One-way and two-way ANOVA and multiple comparison tests (Dunnett, Tukey, and Sidak) were used to investigate differences between experimental groups using GraphPad Prism 8.0 software (USA). *p* < 0.05 was considered statistically significant.

## 4. Conclusions and Future Perspectives

In conclusion, FLt showed antitumor effects in vitro by inducing both intrinsic and extrinsic apoptosis and strongly inhibiting the transcriptional expression of Bcl-2 and XIAP, which are known for their antiapoptotic activity. FLt activated effector caspase 3/7 and induced caspase 8, caspase 9, and AIF transcripts associated with proapoptotic activity. In addition, it is suggested that AIF exerts necrotic activity. Further research is needed to fully understand the apoptotic effects, which could clarify and support the hypothesis of anticancer mechanisms of FLt. These preliminary results suggest that FLt is a promising candidate for the adjuvant treatment of TNBC. This discovery adds value to the algal bioresources found in the Peruvian sea.

## Data Availability

Data are contained within this article.

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
