# Peer review of "Fucoidan from Lessonia trabeculata Induces Apoptosis through Caspase Dependent and Caspase-Independent Activation in 4T1 Breast Adenocarcinoma In Vitro"

_marinedrugs, 2024, doi:10.3390/md22060251_

Round 1

Reviewer 1 Report

Comments and Suggestions for Authors

This study involved inducing apoptosis in 4T1, a breast cancer cell line, using fucoidan extracted from S.A. By confirming the apoptosis signaling system of 4T1 cells, the authors reached the conclusion that fucoidan can induce apoptosis. However, the research content is very poor and unprofessional. In particular, too many experiments were not performed in the process of proving apoptosis. As a result, it is difficult for the experimental results to sufficiently support the conclusion.

1.     The paper was not written in a standard format. There is too much discussion in the results section. And this dicussion describes content that cannot be resolved in this discussion. For example, this study moves on to another experiment without any mention of how the content in lines 80-82 will be explained.

2.     Morphological change analysis of apoptosis should be added.

3.     Monosaccharide composition is required to determine whether the fucoidan used is really fucoidan.

4.     Annexin-V only reacts with binding buffer, so I don't know how the experiment was conducted under PBS (+FBS) conditions?

5.     Why does DOX appear in the fucoidan extraction in Method section 3.3?

6.     They keep mentioning IC50, but just write down the treatment concentration.

7.     7-AAD cannot be used together with DOX for fluorescence measurement at the same wavelength. How was it measured?

8.     DNA fragmentation results are too unclear. The DNA does not appear to be broken.

9.     In the Annexin-V experiment results, compensation was not applied, so the FACS results looked strange.

10.  Usually, the negative control group, which is the figures in the paper, appears first, and the positive control group appears the last. The negative control group appears the last, making it difficult to interpret the results.

11.  The results of stimulating mitochondria are insufficient. Analysis methods such as mitochondrial damage analysis, cytochrome c release, and Apaf-1 should be added.

12.  The signaling system of apoptosis uses western blot as a basis. Additional analysis is required for the activity of proteins that need to be analyzed, such as caspase-3, 8, 9, and PARP.

Author Response

Dear Reviewer:

Reviewer 2 Report

Comments and Suggestions for Authors

In this manuscriptFucoidan from Lessonia trabeculata induces apoptosis through caspase dependent and caspase-independent activation in 4T1 breast adenocarcinoma in vitro” , The authors in order to obtain a comprehensive understanding of the levels of biological activity and cytotoxicity. In vitro cytotoxicity assays are performed using MTT viability assays, annexin V, and caspases 3/7-mediated apoptosis. In addition, DNA fragmentation and transcriptional expression of anti-apoptotic and pro-apoptotic genes were analyzed in TNBC 4T1 cells. In addition, the FLt + Dox assay was performed. Current studies have shown caspase-dependent and caspase-independent mechanisms for FLt-induced apoptosis. These findings suggest that FLt may have potential applications in breast cancer treatment and as an inflammatory modulator, a finding that increases the value of Peruvian seaweed bioresources. However, there had some problems and questions, the comments were as followed:

1.     When MTT first appeared, the author was advised not to abbreviate.

2.     It is recommended that annexin V be described in the abstract rather than in 2.2.

3.     What does "inactive complex of chaperone ICAD" on line 129 mean?

4.     What lines 309-315 mean, the author is advised to check it himself.

5.     Is "L. trabeculata" in line 324 a species, why can it be confirmed, and is there any correlation between them?

6.     Whether the format of the references is correct, whether the year needs to be bolded, whether others need to be italicized, etc., it is recommended that the author carefully check and modify the format.

7.     The materials and methods section also did not see the authors grouping the experiments.

Author Response

Dear Reviewer:

Round 2

Reviewer 1 Report

Comments and Suggestions for Authors

The author is completely ignoring the reviewer's comments. This reviewer has been conducting research related to biology and apoptosis for over 20 years, so I selected and presented only the truly necessary advice, but it was completely ignored by the author. It is judged that there is no need to further review this paper.